# DNA-PK is a DNA sensor for IRF-3-dependent innate immunity

**Brian J Ferguson[1,2‡], Daniel S Mansur[1†‡], Nicholas E Peters[1‡], Hongwei Ren[1,2], Geoffrey L Smith[1,2]***

[1]Department of Virology, Imperial College London, London, United Kingdom; [2]Department of Pathology, University of Cambridge, Cambridge, United Kingdom

**Abstract** Innate immunity is the first immunological defence against pathogens. During virus infection detection of nucleic acids is crucial for the inflammatory response. Here we identify DNA-dependent protein kinase (DNA-PK) as a DNA sensor that activates innate immunity. We show that DNA-PK acts as a pattern recognition receptor, binding cytoplasmic DNA and triggering the transcription of type I interferon (IFN), cytokine and chemokine genes in a manner dependent on IFN regulatory factor 3 (IRF-3), TANK-binding kinase 1 (TBK1) and stimulator of interferon genes (STING). Both cells and mice lacking DNA-PKcs show attenuated cytokine responses to both DNA and DNA viruses but not to RNA or RNA virus infection. DNA-PK has well-established functions in the DNA repair and V(D)J recombination, hence loss of DNA-PK leads to severe combined immunodeficiency (SCID). However, we now define a novel anti-microbial function for DNA-PK, a finding with implications for host defence, vaccine development and autoimmunity.

**\*For correspondence:**
gls37@cam.ac.uk

**†Present address:** Laboratory of Immunobiology, Departmento de Microbiologia, Imunologiae Parasitologia, Universidade Federal de Santa Catarina, Florianopolis, Brazil

**‡**These authors contributed equally to this work

**Competing interests:** The authors have declared that no competing interests exist

**Reviewing editor**: Ruslan Medzhitov, Yale University, United States

## Introduction

The innate immune response is mediated by the production of cytokines, including type I interferons (IFNs), and chemokines following the detection of pathogen-specific molecules by host cells (*Akira et al., 2006*; *Medzhitov, 2007*). Detection of nucleic acids is crucial in triggering the innate immune to pathogens, particularly in response to viruses (*Pichlmair and Reis e Sousa, 2007*). Various double and single stranded RNA substrates are recognised directly by the DExD/H box RNA helicases retinoic acid-inducible gene I (RIG-I) and melanoma differentiation-associated gene-5 (MDA-5) in the cytoplasm and by the endosomal toll-like receptors (TLRs) (*Pichlmair and Reis e Sousa, 2007*). The engagement of such RNA receptors leads to the rapid transcription of genes encoding anti-viral proteins via the activation of transcription factors belonging to the interferon regulatory factor (IRF) and nuclear factor-kappa B (NF-κB) families. It has been recognised for some time that intracellular DNA can activate a similar IRF-3-dependent innate immune response (*Stetson and Medzhitov, 2006*) and it is known that this signalling pathway depends on both the IRF-3-activating kinase, TANK-binding kinase 1 (TBK1) (*Ishii et al., 2006, 2008*) and the adaptor protein stimulator of IFN genes (STING—also known as MITA, ERIS and TMEM173) (*Ishikawa et al., 2009*). Only more recently, however, have some candidate receptors for these pathways been identified. DNA-dependent activator of IFN-regulatory factors (DAI) (*Takaoka et al., 2007*), RNA polymerase III (RNA-Pol III) (*Ablasser et al., 2009*; *Chiu et al., 2009*), IFN inducible gene 16 (*Unterholzner et al., 2010*) and DDX41 (*Zhang et al., 2011b*) have been described as cytoplasmic DNA sensors that activate IRF-3. Nonetheless, the in vivo relevance of these sensors remains unknown and the normal immune response to DNA stimulation of $Dai^{-/-}$ and $Ips1^{-/-}$ cells (*Ishii et al., 2008*; *Wang et al., 2008*) and sensing of plasmodium DNA independent of these receptors (*Sharma et al., 2011*) indicates other IRF-3-activating cytoplasmic DNA sensors exist. In addition, although TBK1 and STING are essential for activation of IRF-3 following DNA stimulation, the molecular details of this signalling pathway are poorly understood (*Paludan et al., 2011*; *Barber, 2011*).

**eLife digest** For multicellular organisms, the innate immune system is the first immunological defence against infection, rapidly recognizing and responding to the presence of any pathogen. Many different cell types contribute to the innate immunity, including fibroblasts, epithelial cells, dendritic cells and macrophages. Once alerted to injury or infection, these cells release proteins called cytokines, interferons and chemokines into the blood or directly into tissue. These proteins act as messengers and interact with receptors on the surfaces of other cells in the immune system, stimulating them to join the battle against the infection.

Detecting nucleic acids such as DNA is an important part of recognizing pathogens and infectious agents, particularly viruses, and activating the innate immune system. However, while the presence of DNA in the cytoplasm is known to initiate an innate immune response, we do not fully understand how this foreign DNA is sensed, or how the innate immune system is activated once foreign DNA has been detected.

Here Ferguson et al. report that a well-known complex of three proteins, collectively called DNA-dependent protein kinase, is able to activate an innate immune response when it detects foreign DNA. This enzyme, called DNA-PK for short, is best known for its ability to repair broken DNA inside the nucleus. Now Ferguson et al. have found that it is also present at high levels within fibroblasts, cells that are often primary targets of viral infection, and they go on to explain how the detection of DNA by DNA-PK triggers a sequence of events that leads to the innate immune response being activated. These events include the transcription of type I interferon, chemokines and cytokines in a manner that depends on the presence IRF-3, a transcription factor that has a central role in the response of the immune system to viral infection.

By identifying a role for DNA-PK in the cytoplasm as a DNA sensor, the work of Ferguson et al. increases our understanding of innate immunity. It may also, in the future, lead to an improved understanding of autoimmunity, and might also assist in the development of more immunogenic vaccines based on DNA or microbes that contain DNA.

The presence of unidentified DNA sensors in fibroblasts is especially pertinent to virus infection since these cells are often a primary target of virus infection in vivo. These cells should have sentinel innate immune receptors in place to detect the presence of foreign nucleic acid and respond by producing IFN, cytokines and chemokines to initiate the anti-viral state in the surrounding tissue as well as to attract immune cells to the site of infection. In this study we identify DNA-dependent protein kinase (DNA-PK) as a novel DNA sensor in fibroblasts where it is present at high levels enabling it to respond to incoming infection without the need for prior stimulation. DNA-PK is a heterotrimeric protein complex consisting of three proteins, Ku70, Ku80 (also known as Ku86) and the catalytic subunit DNA-PKcs (encoded by the *xrcc6*, *xrcc5* and *prkcd* genes respectively). Ku70 and Ku80 themselves form a heterodimer and the absence of one subunit de-stabilises the expression of the other (*Nussenzweig et al., 1996*; *Gu et al., 1997*). Both the Ku heterodimer (*Walker et al., 2001*) and DNA-PKcs (*Hammarsten and Chu, 1998*) can bind directly to DNA but, in the absence of Ku the affinity of DNA-PKcs for DNA is greatly reduced (*Yaneva et al., 1997*). DNA-PK has a well described role in the nucleus where it is necessary for non-homologous end joining (NHEJ) and so has a key role in repairing double-strand DNA breaks (*Lieber et al., 2003*). DNA-PK has also been detected in the cytoplasm by immunofluorescence and cell fractionation (*Huston et al., 2008*; *Balazs et al., 2012*), although prior to this study no function has been assigned to DNA-PK in this localisation. Here we found that, in the cytoplasm, DNA-PK signals via IRF-3 to activate an anti-microbial innate immune response to DNA mediated by the production of IFN, cytokines and chemokines. We show that DNA-PK co-localises with sites of viral DNA replication during VACV infection and the innate immune response to DNA and to infection with vaccinia virus (VACV) and herpes simplex virus (HSV-1) was impaired in both cells and mice which lack components of DNA-PK.

## Results

### DNA-PK binds DNA in the cytoplasm

To identify novel cytoplasmic DNA sensors, we transfected biotinylated dsDNA composed of a con-catenated 45-bp oligonucleotide (called immunostimulatory DNA, ISD, (*Stetson and Medzhitov, 2006*))

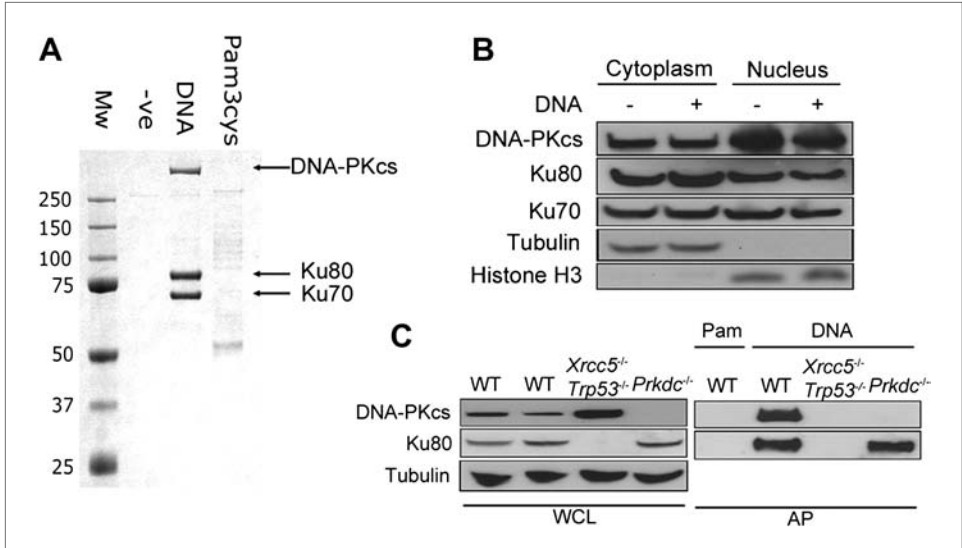

**Figure 1**. DNA-PK binds DNA in the cytoplasm. (**A**) Biotinylated DNA or biotinylated pam-3-cys were transfected into HEK 293T cells and 30 min later DNA was affinity purified from the cytoplasmic fraction before analysis of DNA/protein complexes by SDS-PAGE. The three major protein bands visible by coomassie blue staining were excised from the gel and identified as Ku70, Ku80 and DNA-PKcs by mass spectrometry. (**B**) Cells were untreated or transfected with DNA and 2 hr later proteins were extracted from the nucleus and cytoplasm. 50 µg of protein from each fraction (representing 10% of total cytoplasmic protein by volume and 5% of the total nuclear protein) was analysed by immunoblotting for DNA-PKcs, Ku70 and Ku80. β-tubulin and histone-H3 were used as controls to indicate successful fractionation. (**C**) Ku80 is required for efficient binding to DNA. MEFs of indicated genotypes were transfected with DNA biotinylated at the 3' end or biotinylated pam-3-cys (Pam) and lysed 1 hr later. Following affinity purification (AP) of biotinylated ligands with streptavidin agarose, proteins were analysed by SDS-PAGE and immunoblotting for DNA-PKcs and Ku80. WCL; whole cell lysate.

into human embryonic kidney (HEK) 293T cells and isolated DNA/protein complexes from the cytoplasm by affinity purification (*Figure 1A*). The three abundant proteins that bound specifically to DNA, and not to biotinylated lipoprotein pam-3-cys, were unequivocally identified as Ku70, Ku80 and DNA-PKcs (*Figure 1A*). We confirmed by subcellular fractionation that DNA-PK is present in the cytoplasm of resting cells (*Figure 1B*), as reported previously (*Huston et al., 2008*; *Balazs et al., 2012*) and that this cytoplasmic localisation was not due to nuclear contamination (*Figure 1B*). The DNA pulldown was reproduced from murine fibroblasts (*Figure 1C*), showing the cross-species conservation of this interaction. Additionally, in murine embryonic fibroblasts (MEFs) lacking Ku80 (*Xrcc5$^{-/-}$*), the remaining DNA-PK components did not interact with cytoplasmic DNA under the conditions tested, whereas in *Prkdc$^{-/-}$* MEFs, Ku80 was still recruited (*Figure 1C*). Therefore, the association between DNA-PKcs and cytoplasmic DNA is enhanced by Ku.

## DNA-PK activates the innate immune response to DNA in fibroblasts

In fibroblasts, the transcription of genes encoding chemokines and cytokines was induced in response to DNA in a length and dose-dependent manner, was sensitive to DNase treatment and required DNA to be transfected into the cell (*Figure 2A* and data not shown). These observations are consistent with other studies (*Karayel et al., 2009*; *Unterholzner et al., 2010*). Next we tested DNA from various sources, including vaccinia virus (VACV) and *Escherichia coli*, for its ability to bind DNA-PK in the cytoplasm and to stimulate *Cxcl10* transcription. *Cxcl10* is strongly induced by intracellular nucleic acids (*Ishii et al., 2006*) via the IRF-3- and NF-κB-binding sites in its promoter (*Spurrell et al., 2005*). Each of these different DNA species associated with DNA-PK in MEFs, and this association correlated with *Cxcl10* induction (*Figure 2B,C*—black bars), implying DNA sequence-independence of this response. A similar correlation between DNA binding and *Il6* induction was also observed (data not shown).

Strikingly, in MEFs lacking DNA-PKcs (*Prkdc$^{-/-}$*) there was a significant impairment in *Cxcl10* transcription in response to double stranded concatenated ISD DNA (*Stetson and Medzhitov, 2006*)

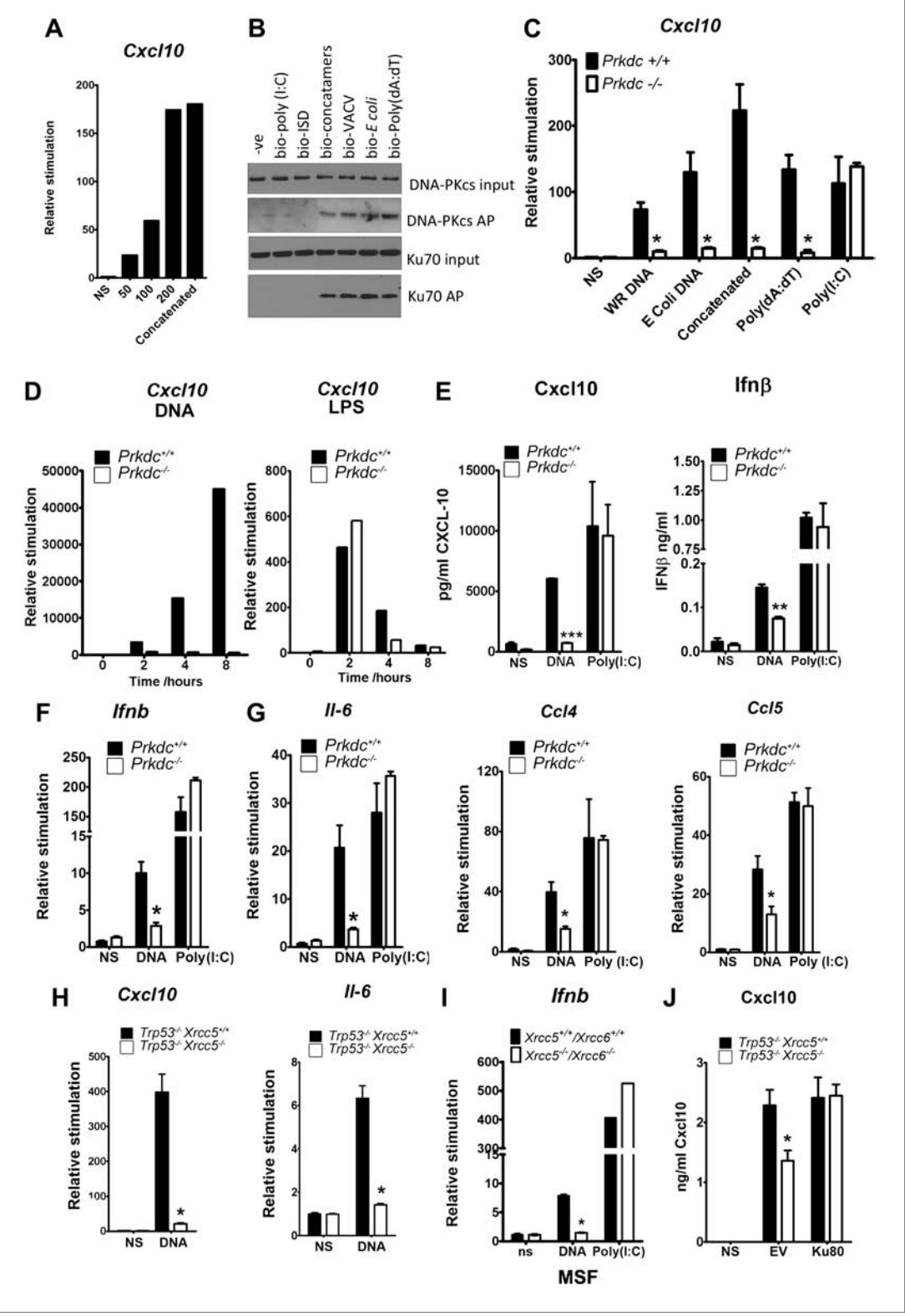

**Figure 2**. The innate immune response to DNA requires DNA-PK in fibroblasts. (**A**) ISD DNA of different lengths was transfected into MEFs and the transcription of *Cxcl10* was assayed by qPCR 6 hr later. (**B**) Double stranded oligonucleotides (bio-ISD), concatenated ISD DNA (bio-concatamers), genomic vaccinia virus DNA (bio-VACV), genomic *E. coli* DNA (bio-*E. coli*), poly (dA:dT) or the RNA analogue poly (I:C) were biotinylated and transfected into HEK293 cells. Following affinity purification of proteins from cytoplasmic extracts using streptavidin

*Figure 2. Continued on next page*

*Figure 2. Continued*

agarose, the bound proteins were analysed by SDS-PAGE and immunoblotting. AP; affinity purification. (**C**) Primary MEFs of the indicated genotype were transfected with 10 µg/ml of the same (non-biotinylated) nucleic acids as in (**A**) followed by qRT-PCR analysis measuring induction of *Cxcl10* mRNA 6 hr later. (**D**) Wild type and *Prkdc*⁻/⁻ transformed MEFs were transfected with DNA (10 µg/ml, left panel) or stimulated with LPS (100 ng/ml, right panel) and the level of transcription of *cxcl10* was measured at the indicated times post stimulation. (**E**) Levels of *Cxcl10* and Ifnβ were measured by ELISA from the supernatants of primary wild type and *Prkdc*⁻/⁻ MEFs at passage 1, 24 hr after transfection with DNA or poly (I:C). (**F**),(**G**) Primary wild type and *Prkdc*⁻/⁻ MEFs at passage 1 were transfected with DNA or poly (I:C) and the level of induction of (**F**) *Ifnb* and *Il-6* and (**G**) *ccl4* and *ccl5* mRNA was measured by qRT-PCR 6 hr later. (**H**) MEFs expressing Ku80 or lacking Ku80 were transfected with DNA and the transcription of *Cxcl10* or *Il6* was measured by qPCR 6 hr later. (**I**) Primary murine skin fibroblasts (MSF) from wild type adult mice or those lacking both Ku genes were transfected with DNA or poly (I:C) and the level of *Ifnb* induction was measured 6 hr later by qRT-PCR. (**J**) *Xrcc5*⁺/⁺/*Trp53*⁻/⁻ and *Xrcc5*⁻/⁻/*Trp53*⁻/⁻ MEFs were transfected with an expression plasmid encoding Ku80 or an empty vector (EV) control and *Cxcl10* production was measured 24 hr later by ELISA. *** p<0.001, ** p<0.01, * p<0.05, n ≥ 3, error bars ± SEM, ns; non-stimulated.

(from now on referred to as DNA), viral and bacterial DNA and poly(dA:dT) (*Figure 2C*, white bars). In contrast, the response to the dsRNA analogue, poly(I:C), was similar in the presence or absence of DNA-PKcs indicating the signalling defect is specific to DNA (*Figure 2C*). Time-course experiments indicated that the defect in DNA stimulation of *Prkdc*⁻/⁻ cells was consistent at all times tested (*Figure 2D*) and that these cells are not simply responding to DNA with different kinetics. The data presented in *Figure 2C,D* were carried out with MEFs derived from separate strains of *Prkdc*⁻/⁻ mice, indicating that this phenotype was not confined to a single MEF line. As a control we also tested the response to LPS (*Figure 2D*, right panel) and found the *Prkdc*⁻/⁻ cells responded like the wild type cells, with typically rapid kinetics, to this stimulus. In addition, the secretion of Cxcl10 and Ifnβ as well as the transcription of *ifnb*, *il6*, and the chemokines *Ccl4* and *Ccl5*, was also consistently impaired in multiple preparations of passage 1 *Prkdc*⁻/⁻ MEFs in response to DNA but not RNA (*Figure 2E–G*). It is notable that, in MEFs, the transcription of other type 1 IFNs, type III IFNs and anti-inflammatory cytokines such as *Il4* and *Il10* was not observed in response to DNA stimulation (data not shown). Defects in the production of cytokines in response to DNA were found in transformed MEFs lacking Ku80 (*Xrcc5*⁻/⁻) (*Figure 2H*) or Ku70 (*Xrcc6*⁻/⁻) (not shown) and the transcription of *Ifnb* in response to DNA was also impaired in primary adult murine skin fibroblasts lacking both Ku genes (*Figure 2I*) indicating that this phenotype requires both Ku and DNA-PKcs and is not restricted to embryonic cells. Finally, the re-expression of Ku80 in *Xrcc5*⁻/⁻ cells restored their DNA-dependent production of Cxcl10 to wild-type levels (*Figure 2J*). Collectively, these data indicate that DNA-PK acts as a DNA sensor by binding foreign DNA in the cytoplasm and activating a host innate immune response.

## DNA-PK activates IRF-3 dependent innate immune responses

We delimitated the signalling pathway downstream of DNA-PK by using various MEFs lacking specific signalling components. In *Irf3*⁻/⁻cells the increase in transcription of *Ifnb*, *Cxcl10*, *Il6* and *Isg54* (an IRF-3-dependent gene (*Navarro et al., 1998*)) in response to DNA was lost (*Figure 3A*). Equally, the up-regulation of *Il6* and *Cxcl10* mRNA was lost in *Tbk1*⁻/⁻ MEFs (*Figure 3B*) and primary *Tmem173*⁻/⁻ MEFs (data not shown). Similar results have been reported previously (*Stetson and Medzhitov, 2006*; *Ishii et al., 2006*; *Ishikawa et al., 2009*) and in a similar manner confirmed that cytoplasmic DNA sensing in fibroblasts is independent of TLRs (*Ishii et al., 2006*), IPS-1 (*Kumar et al., 2006*) (and hence both dsRNA-sensing pathways (*Kumar et al., 2006*) and RNA-pol III-dependent DNA sensing (*Ablasser et al., 2009*; *Chiu et al., 2009*)), and DAI (*Ishii et al., 2008*; *Wang et al., 2008*), because there was no defect in DNA-dependent cytokine production in *Myd88*⁻/⁻/*Ticam1*⁻/⁻, *Mavs*⁻/⁻ or *Zbp1*⁻/⁻ MEFs (data not shown).

To establish whether DNA-PK acts upstream of IRF-3 activation, we monitored IRF-3 translocation in response to DNA and RNA stimulation. In *Prkdc*⁻/⁻ MEFs IRF-3 translocation in response to DNA, but not to RNA, was abrogated (*Figure 4A*) and we obtained similar results in *Xrcc5*⁻/⁻ MEFs (*Figure 4B*). In contrast, translocation into the nucleus of the p65 component of NF-κB, was unaffected by absence of the *Prkdc* gene (*Figure 4C*). To confirm the activity of IRF-3 and NF-κB at the transcriptional

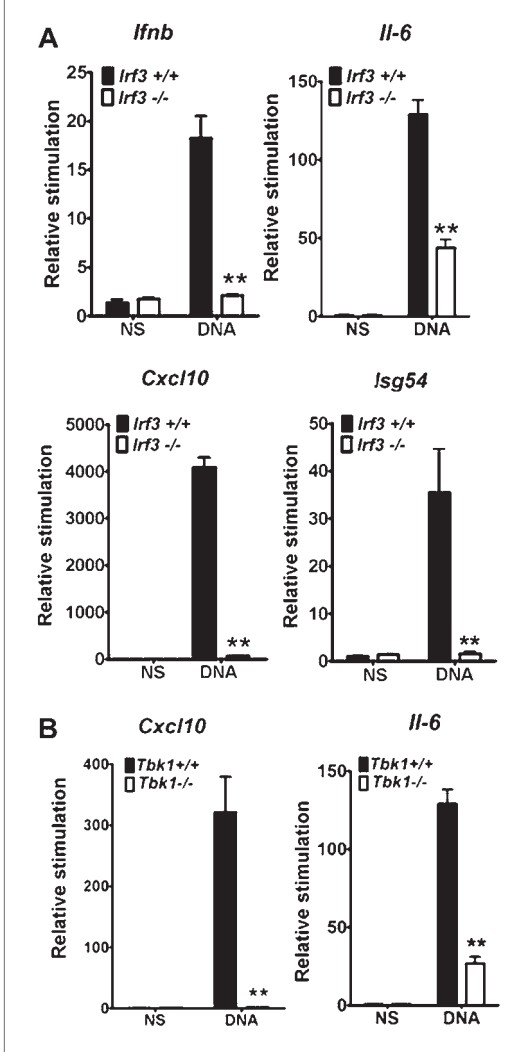

**Figure 3**. IRF-3 and TBK1 are required for the stimulation of multiple genes in response to DNA. (**A**) Primary wild type or *Irf3⁻ᐟ⁻* MEFs were transfected with DNA and the level of induction of *Ifnb*, *Il6*, *Cxcl10* and *Isg54* mRNAs were measured by qRT-PCR 6 hr later. (**B**) Immortalised wild type or *Tbk1⁻ᐟ⁻* MEFs were transfected with DNA or poly(I:C) and the level of induction of *Il-6* and *Cxcl10* was measured by qRT-PCR 6 hr later. (**B**) *** p<0.001, ** p<0.01, n = 3, error bars ± SEM, ns; non-stimulated.

level we examined induction of the IFN-inducible gene *Isg54* (***Navarro et al., 1998***), which is entirely dependent on IRF-3 activity following DNA stimulation (***Figure 3A***). We also assayed expression of *Nfkbia*, a known NF-κB-dependent gene that encodes the IκBα protein (***Rupec et al., 1999***). Induction of *Isg54*, but not *Nfkbia*, was impaired in *Prkdc⁻ᐟ⁻* MEFs in response to DNA but not RNA transfection (***Figure 4D***). These data confirm the DNA-specific defect in IRF-3, but not NF-κB, activation in cells lacking DNA-PKcs and demonstrate that DNA-PK acts as a DNA sensor upstream of the IRF-3-dependent innate immune response. Our data also imply that the existence of an additional DNA sensing pathway in MEFs that is independent of DNA-PK and capable of activating NF-κB.

## DNA-PK activates IRF-3 signalling independent of kinase activity

Previously, DNA-PK was shown to interact directly with IRF-3 (***Karpova et al., 2002***). We confirmed this interaction by co-immunoprecipitation of both proteins (data not shown). In the study of ***Karpova et al. (2002)*** DNA-PKcs was shown to phosphorylate IRF-3 at its N terminus, so enhancing its nuclear retention (***Karpova et al., 2002***). To test whether phosphorylation of IRF-3 is necessary for the DNA-sensing function of DNA-PK, two approaches were taken. First, we tested *Prkdc^{SCID}* MEFs, which express a kinase-dead form of the DNA-PKcs protein lacking its C terminal 83 amino-acids (***Blunt et al., 1996***; ***Guimarães-Costa et al., 2009***), for their ability to produce *Ifnb*, *Cxcl10* and *Il-6* in response to DNA or RNA. However, *Prkdc^{SCID}* MEFs showed no defect in induction of *Ifnb* (***Figure 5A***), *Il-6* (***Figure 5B***), and *Cxcl10* (***Figure 5C***) in response to either DNA or RNA stimulation. Second, we used the DNA-PKcs-specific inhibitor NU7026 (***Veuger et al., 2003***) to inhibit DNA-PKcs kinase activity in wild type MEFs. Cells treated with NU7026 showed no impairment in production of *Cxcl10* in response to DNA transfection (***Figure 5D***). Hence, the kinase activity of DNA-PKcs is not required for activation of IRF-3 in response to DNA.

The finding that the kinase activity of DNA-PKcs was not necessary for DNA sensing prompted an investigation of other interactions with signalling components. To do this we developed an inducible system to study the DNA sensing signalling pathway. HEK293 cells do not activate IRF3 in response to DNA stimulation due to a defect in STING, but by introducing an inducible version of STING into these cells, and then inducing STING expression with doxycycline, the cells activate IRF-3 as shown by its phosphorylation (***Figure 5E***) and produce cytokines (not shown) in response to double stranded ISD DNA concatamers. Using this system, immunoprecipitation showed that prior to DNA stimulation, STING exists in a complex with Ku70 but this interaction is abrogated following transfection of DNA (***Figure 5F***). These data suggest that DNA-PK interacts with the STING-dependent signalling pathway and this changes upon DNA activation.

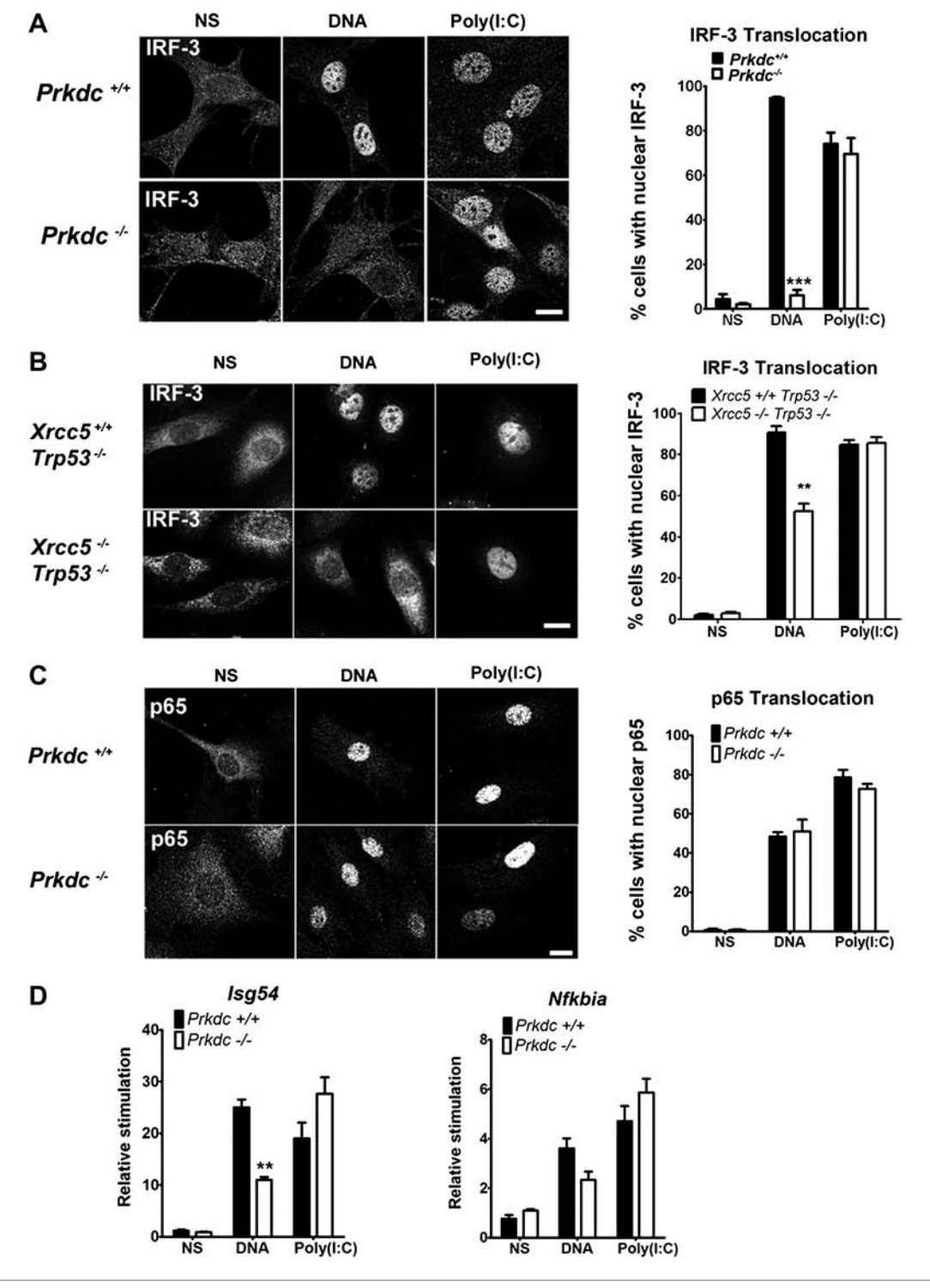

**Figure 4**. DNA-PK activates IRF-3-dependent, NF-κB–independent signalling. (**A**) The localisation of endogenous IRF-3 was analysed by immunofluorescence 1 hr after transfection of primary wild type or *Prkdc*[−/−] MEFs with DNA or poly (I:C) (left panels) and quantified by scoring cells with nuclear staining (right panels, n = 3, counts of at least 50 nuclei per slide in randomised fields of view). (**B**) As (**A**) but with *Xrcc5*[+/+]/*Trp53*[−/−] and *Xrcc5*[−/−]/*Trp53*[−/−] MEFs. (**C**) Analysis of p65 translocation in wild type and *Prkdc*[−/−] MEFs carried out as in (**A**). (**D**) Primary wild type or *Prkdc*[−/−] MEFs were transfected with DNA or poly (I:C) and the level of induction of *Isg54* and *Nfkbia* were measured by qRT-PCR 6 hr later. ** p<0.01, * p<0.05, n = 3, error bars ± SEM. ns; non-stimulated. Scale bar; 10 µm.

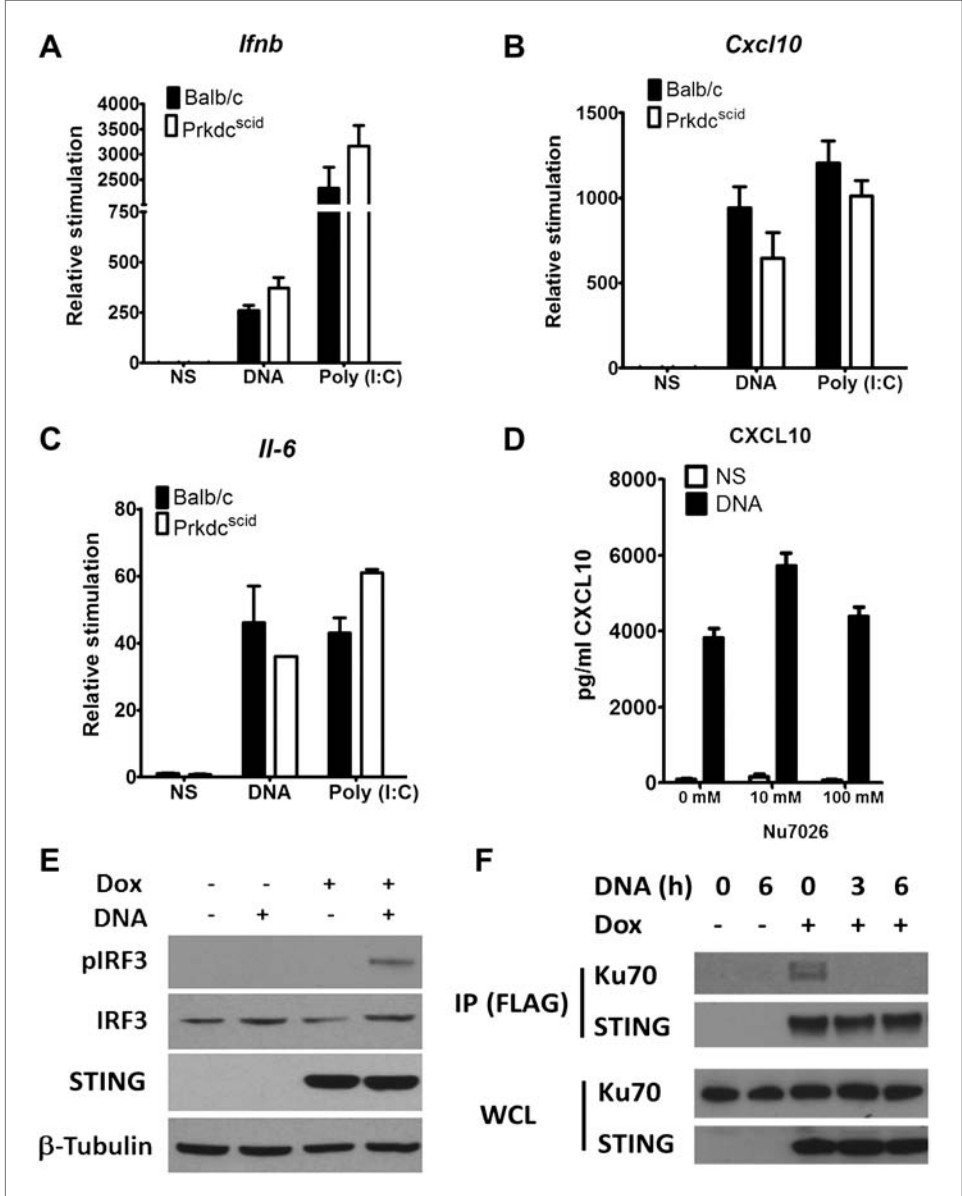

**Figure 5**. DNA-PKcs kinase activity is dispensable for the innate immune response to DNA. Primary fibroblasts from Balb/c or *Prkdc^SCID* mice were transfected with DNA or poly(I:C) or infected with MVA or NDV and the level of induction of (**A**) *Ifnb*, (**B**) *Cxcl10*, or (**C**) *Il-6* was measured 6 hr later by qRT-PCR. (**D**) Fibroblasts were incubated with the indicated dose of DNA-PKcs kinase inhibitor, Nu7026, or carrier control and then stimulated with 10 µg/ml DNA. *Cxcl10* was measured by ELISA in the supernatants 24 hr following stimulation. n = 3, error bars ± SEM. (**E**). Hek293 Trex cells were stably transfected with FLAG-tagged STING under the control of a doxycycline-inducible promoter. STING expression was induced by addition of doxycycline (Dox, 2 µg/ml) for 24 hr and cells were stimulated by transfection with 5 µg/ml DNA for 6 hr. Protein lysates were then immunoblotted with the indicated antibodies. (**F**) STING-293Trex cells were induced to express STING by addition of doxycycline (Dox, 2 µg/ml) for 24 hr and stimulated with 5 µg/ml DNA for the indicated times. STING was then immunoprecipitated and whole cell lystes (WCL) or precipitated proteins (IP) were immunoblotted using the indicated antibodies.

## DNA-PK contributes to the innate immune response to VACV in MEFs

We tested the relevance of this novel DNA-PK-dependent DNA sensing mechanism to virus infection using VACV strain modified virus Ankara (MVA). MVA activates innate immunity via TLR-dependent and independent pathways (*Delaloye et al., 2009*) and, during infection, VACV DNA accumulates in

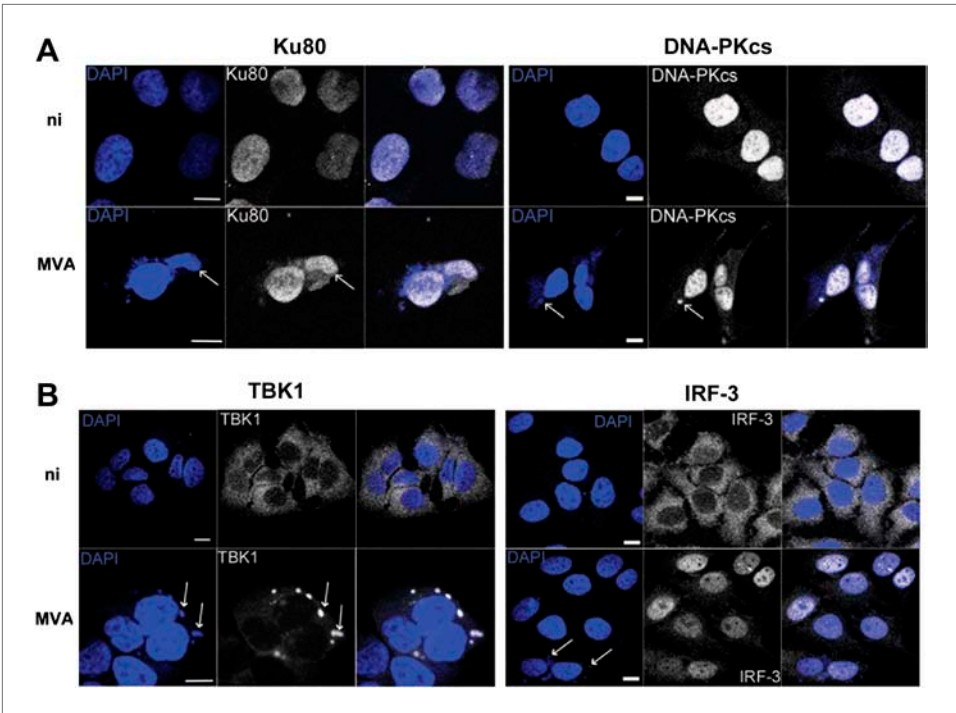

**Figure 6**. DNA-PK and TBK1 localise to sites of VACV DNA replication in infected cells. HeLa cells were untreated or infected with MVA (m.o.i. = 5) for 6 hr. Cells were then fixed and stained with antibodies against (**A**) Ku80 or DNA-PKcs, and (**B**) TBK1 or IRF-3. Cytoplasmic viral factories formed after MVA infection are visualised with DAPI (blue). Sites of co-localisation of DNA-PKcs or Ku80 with viral factories are indicated by white arrows. Scale bars; 10 μm. ni; non-infected.

cytoplasmic virus factories in association with many virus proteins (***Moss, 2007***). We reasoned that such large aggregates of foreign DNA, present in a cellular compartment where DNA does not normally reside, would present an excellent target for an innate immune DNA-sensing mechanism. By 6 hr post infection, both Ku80 and DNA-PKcs had accumulated in these viral factories (***Figure 6A***) together with the IRF-3-activating kinase TBK1 (***Figure 6B***, left panel), consistent with its role in this sensing pathway, although IRF-3 was mostly nuclear, reflecting its activation by virus infection (***Figure 6B***, right panel).

In fibroblasts, MVA induces an IRF-3-dependent response (***Figure 7A***) which is independent of TLR signalling, RNA sensing (and hence RNA-pol III-dependent DNA sensing) and DAI (data not shown), indicating that viral genomic DNA is a major target for the host response to VACV infection. We found the production of *Cxcl10* and *Il-6* was strongly impaired in cells lacking *Prkdc*$^{-/-}$ or *Xrcc5*$^{-/-}$ during MVA infection (***Figure 7B,C***) whereas the response of these cells to infection by Newcastle disease virus (NDV, an RNA virus) remained intact (***Figure 7B,C***). *Isg54*, but not *Nfkbia*, transcription was impaired in response to MVA infection in *Prkdc*$^{-/-}$ MEFs, whilst the response of both genes to NDV was equivalent (***Figure 7D***), directly indicating that the DNA-PKcs-dependent activation of IRF-3 is important in the response to DNA virus infection. Additionally, in primary *Prkdc*$^{-/-}$ MEFs, there was greater MVA protein synthesis 4–8 hr post infection (***Figure 7E***), not only re-enforcing the function of DNA-PKcs in the anti-viral response but also showing the failure to respond to DNA in *Prkdc*$^{-/-}$ MEFs is not due to a failure to infect these cells or synthesise viral macromolecules. These data provide direct evidence that the innate immune response to infection by a DNA virus is regulated by DNA-PK and confirm the role of this complex in cytoplasmic DNA sensing in the context of infection.

## DNA-PK activates the innate immune response to DNA in vivo

To confirm that DNA-PK-dependent DNA sensing contributes to the innate immune response in vivo, we transfected nucleic acids directly into the ear pinnae of mice and assayed the induction of

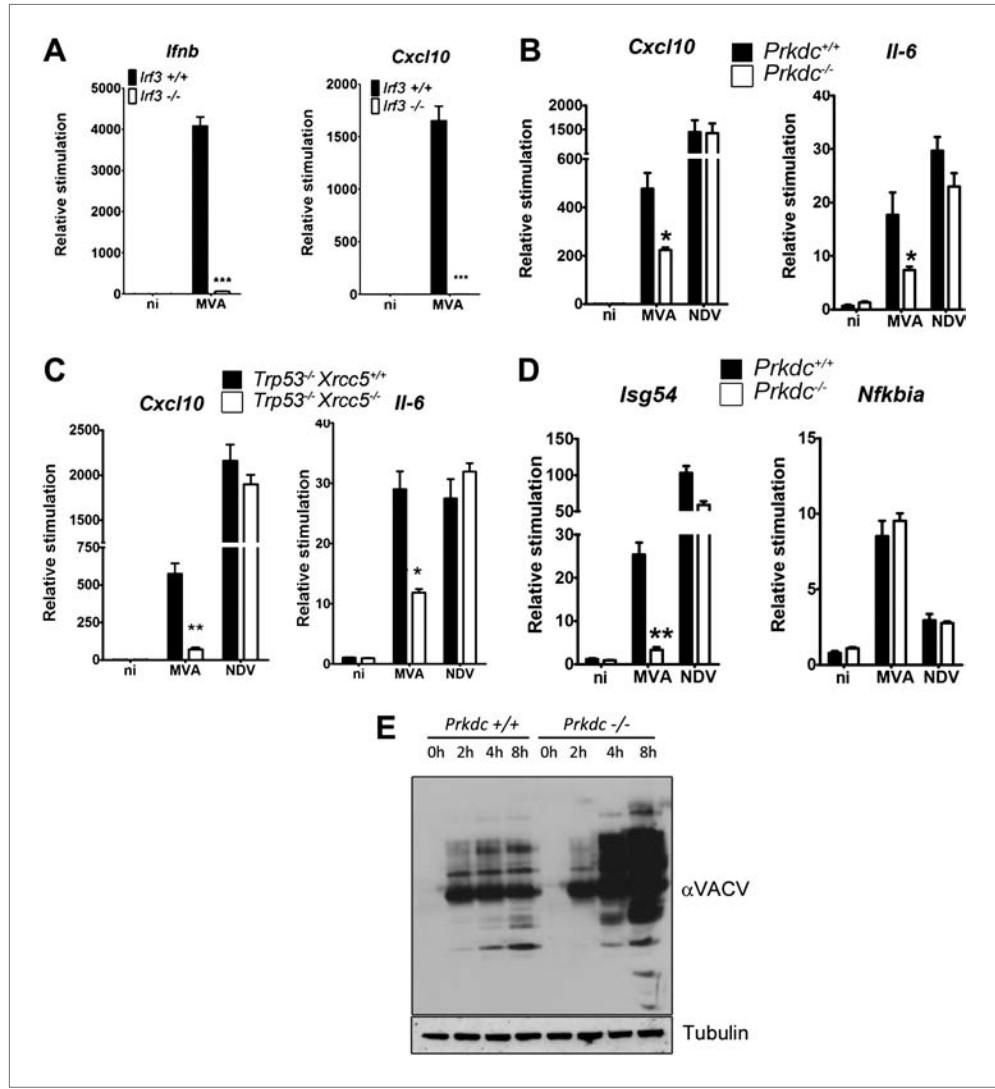

**Figure 7**. DNA-PK contributes to the IRF-3-dependent innate immune response to MVA. (**A**) *Ifnb* and *Cxcl10* transcription was measured 6 hr following MVA infection of primary WT and *Irf3*−/− fibroblasts at an m.o.i. of 5. (**B**) Wild type and *Prkdc*−/− MEFs were infected with MVA or NDV and the level of induction of *Cxcl10* and *Il-6* were measured 6 hr later by qRT-PCR. (**C**) As (**A**) but with immortalised *Xrcc5*+/+/*Trp53*−/− and *Xrcc5*−/−/*Trp53*−/− MEFs. (**D**) The induction of *Isg54* and *Nfkbia* mRNA was measured by qRT-PCR 6 hr after MVA or NDV infection of wild type and *Prkdc*−/− cells. (**E**) Expression of VACV proteins, analysed by immunoblotting with a rabbit polyclonal anti-VACV serum, at the indicated times following infection of primary *Prkdc*+/+ and *Prkdc*−/− MEFs with MVA (m.o.i. = 5). *** $p<0.001$, ** $p<0.01$, * $p<0.05$, n = 3, error bars ± SEM, ni; non-infected.

innate immune transcriptional responses by qPCR analysis of extracted RNA 12 hr later. DNA transfection induced both *Ifnb* and *Il6* and this was significantly reduced in *Prkdc*−/− mice. By contrast, both WT and *Prkdc*−/− mice exhibited equivalent responses to poly(I:C) (**Figure 8A**). Hence, DNA-PKcs plays a key role in potentiating the innate immune response to ectopic DNA in vivo. We next infected WT and *Prkdc*−/− mice with either the DNA virus MVA or with influenza virus (an RNA virus), and assayed expression of *Ifnb* and *Il6* 12 hr later. The absence of DNA-PK severely impaired induction of *Ifnb* and *Il6* in response to MVA but not to influenza virus (**Figure 8B**). Similarly, *Prkdc*−/− mice exhibited severe impairment of *Il6* induction in response to HSV-1 (**Figure 8C**): no *Ifnb* induction could be detected in this experiment. Overall these data confirm that the innate immune response to DNA viruses immediately after infection is significantly dependent upon DNA-PK, despite the presence of other DNA sensors reported hitherto.

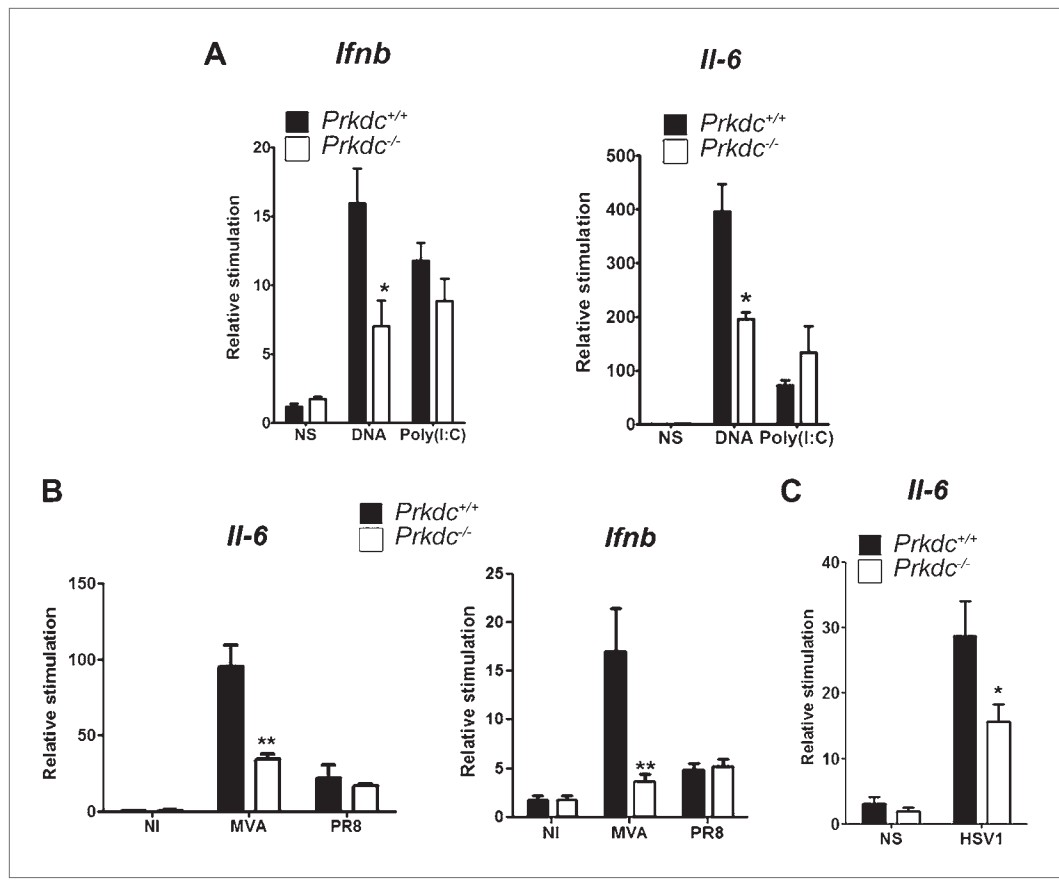

**Figure 8**. DNA-PKcs contributes to the innate immune response to MVA, HSV-1 and DNA in vivo. (**A**) Groups of five *Prkdc*[+/+] and *Prkdc*[−/−] mice were injected intradermally into the ear pinna with cationic lipids complexed with 1 μg DNA or poly(I:C). RNA was extracted from the tissue 12 hr later and *Ifnb* and *Il6* transcription was measured by qPCR. (**B**) Groups of five *Prkdc*[+/+] and *Prkdc*[−/−] mice were infected intradermally with 10[6] pfu of MVA or influenza virus strain A/PR/8/34 (PR8) and 12 hr later *Ifnb* and *Il6* transcription was measured by qPCR from RNA extracted from the local site of infection. (**C**) As (**B**) but with HSV-1 strain S17. Note that levels of *ifnb* could not be measured above background in this experiment. *p<0.1, ** p<0.01, n = 5, error bars ± SEM, ni; non-infected.

## Discussion

The identification of DNA-PK as a DNA sensor advances understanding of the innate immune response to infection and expands the current repertoire of DNA sensing mechanisms. In this study we show that the three proteins which constitute the DNA-PK complex; Ku70, Ku80 and DNA-PKcs, bind in significant amounts to DNA transfected into the cytoplasm of resting cells leading to an IRF-3-dependent innate immune response. Consistent with this, we found DNA-PK and TBK1 localised to sites of DNA replication during virus infection. Previous studies have ruled out a role for DNA-PK in the production of *Ifnb* in response to DNA in monocytic cells (*Stetson and Medzhitov, 2006*) and HEK293 cells (*Zhang et al., 2011a*). However, although DNA-PK components are expressed at high levels in a wide range of tissues and cell types, they are absent in primary macrophages ((*Moll et al., 1999*) and data not shown), and we now show that in fibroblasts and in mice the innate immune response to DNA and DNA viruses is dependent on DNA-PKcs. The in vivo deficiency in DNA sensing in the absence of DNA-PKcs is observed despite the presumed presence of other DNA sensors. The observation that this defect is not extended to RNA, LPS or to RNA viruses shows that DNA-PK loss does not confer a general defect on intracellular innate immune signalling, and the same is true in cultured *Prkdc*[−/−] MEFs which respond normally to RNA. It was noticed that the defect in the innate immune response to DNA was greater in cells lacking DNA-PKcs than in cells lacking Ku, even though Ku can still bind DNA in the absence of DNA-PKcs. This suggests that signalling can progress with

DNA-PKcs interacting with DNA in the absence of Ku but that Ku enhances the signalling process by increasing the affinity of the protein complex for DNA. The phenotypic defect in DNA sensing was consistent in cells from four separate genotypes, ruling out the possibility that it was caused by a second site mutation. Consistent with this reintroduction of Ku80 into *Xrcc5*$^{-/-}$ cells restored DNA sensing (*Figure 2J*).

The well-studied *Prkdc*$^{SCID}$ mutation kills the kinase activity of DNA-PKcs via the introduction of a premature stop codon that results in expression of a truncated protein (*Blunt et al., 1996*). Although kinase activity is essential for DNA-PKcs to function in DNA repair, and therefore V(D)J recombination, this mutation did not affect the ability of DNA-PK to function as an innate immune DNA sensor. The functional significance of the interaction of DNA-PK with IRF-3 and its subsequent N-terminal phosphorylation reported by *Karpova et al. (2002)* remains unknown. However, we show here that Ku interacts with STING in resting cells and that this interaction is abrogated upon DNA stimulation (*Figure 5F*). In the future, further work is necessary to understand how TBK-1 and STING contribute to the full activation of IRF-3 following binding of DNA to DNA-PK to allow IRF-3 translocation to occur.

DNA can act as a powerful immunostimulatory agent in many contexts. DNA vaccination relies on DNA sensing to invoke a powerful innate immune response that, in turn, assists the adaptive response (*Ishii et al., 2008*). Understanding how to optimise such vaccines, therefore, relies on understanding the mechanisms of detection of DNA by the immune system. Furthermore, one of the most common vaccine adjuvants, alum, acts by stimulating the release of DNA from neutrophils (*Zhang et al., 2011b*). Neutrophil extracellular traps (NETs) consist of webs of DNA with globular proteins, are released by a specific form of cell death (*Fuchs et al., 2007*) and function as antimicrobial traps, thereby contributing to the innate immune response (*Brinkmann et al., 2004*).

Outside the context of infection, DNA can act as a damage-associated molecular pattern (DAMP), accelerating inflammatory responses following its release from dying or damaged cells directly contributing to the pathogenesis of various diseases such as atherosclerosis (*Oka et al., 2012*) and deep vein thrombosis (*Brill et al., 2012*). The ability of DNA to act as a DAMP may also link nucleic acid sensing to several autoimmune disorders. In general, autoimmune conditions are characterised by immune responses against host molecules and tissues and are frequently associated with inflammation. Although the causative factors of many such conditions are incompletely understood, it is clear that deregulation of immune signalling may lead to autoinflammation and autoimmunity in some instances (*Rioux and Abbas, 2005*). Indeed, the accumulation of DNA and its subsequent detection by DNA sensing pathways can result in the initiation of autoimmune diseases. Mice which lack either of two enzymes responsible for degradation of DNA, 3′ repair exonuclease 1 (Trex1) and DNaseII, develop spontaneous autoimmune disorders associated with the initiation of IRF-3-dependent cytosolic DNA sensing (*Kawane et al., 2006*; *Stetson et al., 2008*; *Okabe et al., 2008*). The absence of DNaseII results in chronic polyarthritis which is thought to be a result of the inflammation caused by a lack of clearance of DNA from macrophages (*Kawane et al., 2006*; *Okabe et al., 2008*). Trex1 deficiency or mutation, on the other hand, is associated with systemic lupus erythematosus (SLE), chilblain lupus and the human disease Aicardi-Goutières syndrome (AGS) (*Lee-Kirsch et al., 2007a*, *2007b*; *Stetson et al., 2008*) and in this case inflammation is initiated from non-haematopoietic cells via a STING-dependent pathway (*Gall et al., 2012*). Interestingly, defects in the clearance of NETs have also been suggested to contribute to the initiation of SLE (*Hakkim et al., 2010*). Furthermore, the up-regulation of the inflammasome-activating DNA sensor absent in melanoma 2 (AIM2) in a mouse model of lupus and in patients with SLE-associated nephritis (*Roberts et al., 2009*; *Kimkong et al., 2009*) and the presence of anti-DNA-PK and anti-RNA-Pol III antibodies in patients with SLE and systemic sclerosis (*Cavazzana et al., 2008*, *2009*) makes the link between DNA sensing and autoimmune disorders worthy of further investigation.

The identification of several candidate DNA sensors in multiple cells types in recent years (*Hornung and Latz, 2010*) suggests the evolution of redundancy in this system. This redundancy is not surprising given the tendency of pathogens to evolve escape-mechanisms for evading host immune mediators (*Versteeg and García-Sastre, 2010*; *Bardoel and Strijp, 2011*) in turn inducing the host to evolve further pathogen recognition mechanisms. In the context of DNA sensing this is exemplified by the relatively recent evolution of the PYHIN domain proteins, such AIM2 and IFI16 (*Schattgen and Fitzgerald, 2011*), which are a mammalian addition to the ancient innate immune system. Furthermore, a function of DAI was recently uncovered by identifying murine cytomegalovirus protein, vIRA, which

interacts with DAI and inhibits its ability to initiate DNA-induced necroptosis (*Upton et al., 2012*). The biological function for DAI may therefore be in the initiation of a cell death pathway, rather than an IRF-3 dependent inflammatory response. This indicates there are at least three outcomes to cytosolic DNA sensing, the induction of cytokine expression via IRF-3 and NF-kB activation, the secretion of IL-1β via the AIM 2 inflammasome and the induction of necroptosis by DAI. What is not clear though, is how these different responses contribute to the overall immune response to infection by DNA pathogens and to what extent they are cell and tissue-type dependent. Further work is necessary to uncover the relative contributions of these different DNA sensing mechanisms in specific cell types and to different DNA structures as well as to understand how these sensors co-ordinate with STING and other adaptor proteins to activate TBK1 and IRF-3 (*Paludan et al., 2011*; *Barber, 2011*).

Overall these findings provide a novel function for DNA-PK in the innate immune response, beyond its roles in DNA repair and V(D)J recombination, and increase our understanding of the innate immune response to cytoplasmic DNA.

## Materials and methods

### Mice

*Prkdc*$^{+/-}$ mice on a 129 background were a kind gift from Dr Fred Alt (*Gao et al., 1998*) and *Prkdc*$^{SCID}$ and Balb/c mice were from Harlan laboratories. Animals were maintained as required under UK Home Office regulations. Groups of 5 age and sex matched mice were injected intradermally with 1 µg DNA or poly(I:C) (Invivogen, San Diego, CA) pre-incubated with 2 µl Lipofectamine2000 in Optimem (Life Technologies, Grand Island, NY) or injected intradermally with $10^6$ plaque forming units (pfu) of MVA, HSV-1 or influenza virus A/PR/8/34 in PBS. Primary mouse embryonic fibroblasts (MEFs) were isolated from E13.5 embryos derived from time-mated pregnant mice using standard protocols.

### Cell culture and transfection

HEK293T and HEK293 Trex cells (Life Technologies, Grand Island, NY) were maintained in DMEM containing 10% FBS with the addition of blasticidin (5 µg/ml) and zeocin (100 µg/ml) for the selection and maintenance of the inducible STING-expressing cell line (STING-293Trex). MEFs and murine skin fibroblasts (MSFs) from various genetic backgrounds were maintained in DMEM containing 15% FBS, 100 U/ml penicillin and 100 µg/ml streptomycin. Primary *Prkdc*$^{+/+}$ and *Prkdc*$^{-/-}$ MEFs were prepared in house on multiple occasions or supplied as a kind gift by Dr Brian Hemmings and were used only at passage 1. Transformed *Prkdc*$^{-/-}$ MEFs (used solely for experiments leading to the data presented in *Figure 2D*) were a kind gift from Professor Penelope Jeggo. HeLa cells were maintained in RPM.I containing 10% FBS and 2 mM L-glutamine. Transfections were carried out with Fugene6 (Roche, Penzburg, Germany).

### DNA concatenation

Double stranded oligonucleotide DNA, ISD (sense sequence, TACAGATCTACTAGTGATCTATGAC TGATCTGTACATGATCTACA) was phosphorylated at the 5′ end by incubation with T4 polynucleotide kinase (New England Biolabs, Ipswich, MA) for 30 min at 37°C and then ligated with T4 DNA ligase (Promega, Madison, WI) for 16 hr at 15°C. Concatenation was confirmed by agarose gel electrophoresis. This DNA was used at 10 µg/ml for transfection unless stated otherwise.

### Biotinylation

3′ biotinylated oligonucleotides were purchased from IDT DNA Technologies. Other nucleic acids were biotinylated using the Photoprobe biotinylation kit (Vector Labs, Burlingame, CA) following the manufacturer's instructions.

### DNA pull down assay

Concatenated oligonucleotide DNA (sequence as above) that was biotinylated at the 3′ end was transfected into cells using PEI (Sigma-Aldrich, St Louis, MI). After 30 min, cells were lysed in buffer containing 10 mM Tris–Cl, pH 8, 0.1% NP40, 10 mM MgCl$_2$ and the cytoplasmic fraction was isolated by centrifugation at 1500*g* for 3 min. Streptavidin agarose (Thermo Scientific, Rockford, IL), 30 µl, was incubated with the lysate for 1 hr at 4°C and then washed three times in PBS. Purified proteins were analysed by SDS-PAGE and immunoblotting or stained by coomassie-blue and identified by liquid chromatography and tandem mass spectrometry (LC-MS/MS) at the Centre for Systems Biology at Imperial College London.

## Virus infection

VACV strain MVA was purified from cytoplasmic extracts of infected BHK-21 cells by sedimentation through a cushion of 36% (wt/vol) sucrose and was titrated by plaque assay on chicken embryo fibroblasts. NDV and influenza virus strain A/PR/8/34 were kind gifts from Prof Wendy Barclay. HSV-1 strain S17, a gift from Dr Colin Crump, was grown in Vero cells and purified on ficoll gradients. These viruses were used for infections for the indicated times and at indicated doses.

## Immunodetection and immunoprecipitation

For immunoblotting, cell lysates were separated by polyacrylamide gel electrophoresis and transferred onto Immobilon P membranes (GE Healthcare, Little Chalfont, UK). The membranes were blocked in 5% non-fat milk in TBS containing 0.1% Tween 20 for 1 hr at room temperature. Membranes were probed with antibodies against Ku70 (Abcam, Cambridge, UK), Ku80 (Santa Cruz Biotech, Santa Cruz, CA), DNA-PKcs (Millipore, Billerica, MA), tubulin (Millipore, Billerica, MA), histone H3 (Millipore, Billerica, MA), hIRF-3 (Santa Cruz Biotech, Santa Cruz, CA), p-IRF-3 (serine 396, Abcam, Cambridge, UK), HMGB1 (Abcam, Cambridge, UK), FLAG (Sigma-Aldrich, St Louis, MI) or VACV strain Western Reserve (*Law et al., 2006*) and bound immunoglobulin was detected with horse-radish peroxidase-linked secondary antibodies (Agilent, Santa Clara, CA). Ku80, IRF-3 (Life Technologies, Grand Island, NY) or control antibodies were used for immunoprecipitation from HeLa cell lysates expressing IRF-3. FLAG-agarose matrix (Sigma-Aldrich, St Louis, MI) was used for immunoprecipitation of STING from STING-293Trex cells. For immunofluorescence, cells were seeded onto 15 mm glass coverslips, infected with MVA at 5 pfu per cell for 6 hr or transfected and fixed with 4% paraformaldehyde. Cells were permeabilised with PBS containing 0.2% Triton-X100 and blocked with 5% non-fat milk in PBS containing 0.1% Tween 20 for 1 hr at 20°C. Incubation with primary antibodies against Ku70 (Abcam, Cambridge, UK), DNA-PKcs (Millipore, Billerica, MA), mIRF-3 (Life Technologies, Grand Island, NY) hIRF-3, p65 (both Santa Cruz Biotech, Santa Cruz, CA) or TBK1 (Cell Signalling, Danvers, MA) diluted in PBS with 1% milk, for 1 hr at 20°C was followed by detection with alexa-fluor-conjugated secondary antibodies (Life Technologies, Grand Island, NY). Cells were counterstained with DAPI and mounted with Mowiol. Images were obtained with a Zeiss Pascal 510 confocal microscope and processed with Zeiss LSM software (Zeiss, Oberkochen, Germany). For quantification of translocation or IRF-3 or NF-κB into the nucleus 50 cells were counted in random fields of view, in biological triplicates, for each condition and scored for the presence of nuclear staining.

## Enzyme-linked immunosorbent assay (ELISA)

Levels of *Cxcl10* and Ifnβ in cell supernatants were measured using ELISA kits (R&D systems, Minneapolis, MN or PBL, Piscataway, NJ, respectively) according to the manufacturer's instructions.

## Transcriptional analysis

Total cellular RNA was extracted using an RNeasy kit (Qiagen, Hilden, Germany). cDNA synthesis was carried out with Superscript III Reverse Transcriptase (Life Technologies, Grand Island, NY) using 500 ng of template RNA. qPCR was performed on a 7900HT series thermocycler (Life Technologies, Grand Island, NY) with Fast SYBR Green Master Mix (Life Technologies, Grand Island, NY). HPRT was used as the reference gene in all assays. Data were analysed with RQ manager 1.2 software (Life Technologies, Grand Island, NY) and presented as a fold increase relative to time zero. Primers for qPCR were as follows:

> *Cxcl10* For 5′ ACTGCATCCATATCGATGAC 3′,
> *Cxcl10* Rev 5′ TTCATCGTGGCAATGATCTC 3′,
> Ifnβ For 5′ CATCAACTATAAGCAGCTCCA 3′,
> Ifnβ Rev 5′ TTCAAGTGGAGAGCAGTTGAG 3′
> *Ccl5* For 5′ ACGTCAAGGAGTATTTCTACAC 3′,
> *Ccl5* Rev 5′ GATGTATTCTTGAACCCACT 3′,
> *Il-6* For 5′ GTAGCTATGGTACTCCAGAAGAC 3′,
> *Il-6* Rev 5′ GTAGCTATGGTACTCCAGAAGAC 3′,
> Cxcl2 For 5′ GAGCTTGAGTGTGACGCCCCC 3′,
> Cxcl2 Rev 5′ GTTAGCCTTGCCTTTGTTCAG 3′,
> Ccl3 For 5′ ACTGCCTGCTGCTTCTCCTA 3′,
> Ccl3 Rev 5′ TTGGAGTCAGCGCAGATCTG 3′,
> *Ccl4* For 5′ GCCCTCTCTCTCCTCTTGCT 3′,

*Ccl4* Rev 5' CTGGTCTCATAGTAATCCATC 3',
Ccl2 For 5' CTTCTGGGCCTGCTGTTCA 3',
Ccl2 Rev 5' CCAGCCTACTCATTGGGATCA3',
Ifnγ For 5' TCAAGTGGCATAGATGTGGAAGAA3'
Ifnγ Rev 5' TGGCTCTGCAGGATTTTCATG 3',
Il-4 For 5' CATGCACGGAGATGGATG 3',
Il-4 Rev 5' ACCTTGGAAGCCCTACAGAC 3',
Il-10 For 5' TCCTTAATGCAGGACTTTAAGGGTTACTTG 3',
Il-10 Rev 5' GACACCTTGGTCTTGGAGCTTATTAAAATC 3',
HPRT For 5' GTTGGATACAGGCCAGACTTTGTTG 3',
HPRT Rev 5' GATTCAACTTGCGCTCATCTTAGGC 3',
*Nfkbia* For 5' CTGCAGGCCACCAACTACAA3',
*Nfkbia* Rev 5' CAGCACCCAAAGTCACCAAGT 3'
Isg54 For 5' ATGAAGACGGTGCTGAATACTAGTGA 3'
Isg54 Rev 5' TGGTGAGGGCTTTCTTTTTCC 3'

## Statistical analysis

Statistical analysis was carried out using student's t-test with Welch's correction where necessary.

## Acknowledgements

The authors thank Fred Alt for *Prkdc$^{+/-}$* mice, Paul Hasty for supplying *Xrcc5$^{-/-}$/Trp53$^{-/-}$* MEFs and *Xrcc5$^{-/-}$/Xrcc6$^{-/-}$* murine skin fibroblasts, Penelope Jeggo and Brian Hemmings for *Prkdc$^{-/-}$* MEFs, Shiuzu Akira for *Dai$^{-/-}$* and *Myd88/Trif$^{-/-}$* MEFs, Felix Randow for *Tbk1$^{-/-}$* MEFs, Glen Barber for *Sting$^{-/-}$* MEFs, Kate Fitzgerald for *Irf3$^{-/-}$* and *Mavs$^{-/-}$* MEFs, Wendy Barclay for NDV and influenza strain A/PR/8/34 and Colin Crump for HSV-1 strain S17. We thank Christian Ku for the production of the STING-293Trex cell line. Mass spectrometry was carried out by Dr Paul Hitchen at Imperial College Centre for Systems Biology. GLS is a Wellcome Trust Principal Research Fellow.

## Additional information

### Funding

| Funder | Grant reference number | Author |
| --- | --- | --- |
| Wellcome Trust | 090315 | Geoffrey L Smith |
| Medical Research Council | G1000207 | Geoffrey L Smith |
| Medical Research Council | G0800101 | Nicholas E Peters |
| Imperial College London Junior Research Fellowship | JRF2009 | Brian J Ferguson |
| Conselho nacional de Desenvolvimento Cientifico e Tecnologico, CNPq/Brasil | 210239/2006-9 | Daniel S Mansur |

The funders had no role in study design, data collection and interpretation, or the decision to submit the work for publication.

### Author contributions

BJF, Conception and design, Acquisition of data, Analysis and interpretation of data, Drafting or revising the article; DSM, Conception and design, Acquisition of data, Analysis and interpretation of data, Drafting or revising the article; NEP, Conception and design, Acquisition of data, Analysis and interpretation of data, Drafting or revising the article; HR, Acquisition of data, Analysis and interpretation of data; GLS, Conception and design, Analysis and interpretation of data, Drafting or revising the article

### Ethics

Animal experimentation: This work was done with approval from the Home Office of the UK Government under project licence 80/7116.

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
