## [Author Response]

*1) Studies that are based on MEFs can sometimes be misleading due to MEF heterogeneity. Different preps of MEFs are not identical in their gene expression and behavior, and therefore it is critically important A) to test more than one preparation of MEFs to confirm all the key results and B) to perform complementation experiments by re-introducing the gene of interest, in this case DNA-PK, Ku70 and Ku80. It may be technically difficult to transfect DNA-PK because if its size, but Ku70 and Ku80 can be transfected to test recovery of the response to DNA and the virus. These experiments are particularly important because some of the innate immune responses to DNA are not present in transformed cells*.

The importance of using non-transformed (primary) cells was recognised and our study design reflected this. We stated in our submission that *primary* DNA-PKcs ^-/-^ cells were used. With the exception of Figure 2D, all data for DNA-PKcs ^-/-^ cells were from primary (passage 1) cells. In addition, we obtained DNA-PKcs ^-/-^ cells from 2 different external laboratories, as well as making our own primary cells multiple times. All these cells gave the same phenotype. So these data cannot be attributed either to “cell specific artifacts” or due to a “transformed status”.

In addition, the experiments were performed with 4 different mutant cells types with defects in DNA-PK components. These all gave the same defect in DNA-sensing to IRF3. So there is close to zero chance that these observations are due to second site mutations. The signalling defect observed is also highly specific and not general. The mutant cells do not activate IRF3 in response to DNA, but they do respond normally to DNA to activate NF-kB, normally to RNA stimulation, normally to LPS, and normally to RNA virus infection. The suggestion that the defect observed in DNA sensing leading to an IRF3 dependent innate immunity might be due to a general signalling defect is refuted by the data submitted.

*Since transformed MEFs were used in this study, and the degree of transformation may be greater in the knockouts, it is important to perform careful complementation studies*.

The great majority of data used primary (not transformed) cells, and the signalling defect is highly specific rather than general. Therefore, the basis for doing a complementation experiment is lacking. Nevertheless we have done a Ku80 complementation experiment as requested and show that this restores DNA sensing to IRF3 (Figure 2). This is now mentioned in the text. We have also made explicit that the MEFs were from multiple labs including our own and that primary means passage 1.

*2) It is also surprising that the catalytic activity of DNA-PK is not required for the anti-viral response. The authors suggest that DNA-PK may function as a scaffold for signaling complex assembly. However, no evidence is provided here*.

To address this we have created a cell line (based on HEK293 cells) in which STING is expressed inducibly upon addition of doxycycline. HEK293 cells are defective in responding to DNA and this is shown to be overcome by inducible expression of STING (Figure 5E). Using this inducible cell line we show that STING interacts with Ku, and when the pathway is activated by transfection of DNA, the interaction with STING is lost (Figure 5F). This provides additional insight into how DNA-PK components engage the STING / TBK1 / IRF3 pathway, but the full elucidation of how this works is outside the scope of this paper.

*3) Finally, DNA-PK deficiency appears to affect some but not all DNA induced genes. Full gene expression analysis using microarray comparing WT and DNA-PK deficient cells would be very informative. In addition, comparison of gene expression in unstimulated WT and DNA-PK deficient MEFs (which would be part of a control for a microarray study) would help to address the issue of MEF heterogeneity and transformative state*.

Data showing the readout of several inflammatory mediators is presented in this paper. Since DNA-PK is required for IRF3 but not NF-kB activation (as we show), it is evident that some, but not all, DNA-induced genes will be affected, and these are predictable based on the requirement of genes for these different transcription factors. Therefore, we believe that a microarray experiment is unnecessary for this manuscript reporting this novel DNA sensor and showing its in vivo importance.